

# Employing toxin-antitoxin genome markers for identification of *Bifidobacterium* and *Lactobacillus* strains in human metagenomes

Ksenia M. Klimina[1,2], Artem S. Kasianov[1,3], Elena U. Poluektova[1], Kirill V. Emelyanov[3], Viktoriya N. Voroshilova[3], Natalia V. Zakharevich[1], Anna V. Kudryavtseva[4], Vsevolod J. Makeev[1,3,4] and Valery N. Danilenko[1,3]

[1] Vavilov Institute of General Genetics Russian Academy of Sciences, Moscow, Russia
[2] Federal Research and Clinical Center of Physical-Chemical Medicine of Federal Medical Biological Agency, Moscow, Russia
[3] Moscow Institute of Physics and Technology, Dolgoprudny, Russia
[4] Engelhardt Institute of Molecular Biology, Russian Academy of Sciences, Moscow, Russia

Corresponding author
Ksenia M. Klimina,
ppp843@yandex.ru

## ABSTRACT

Recent research has indicated that in addition to the unique genotype each individual may also have a unique microbiota composition. Difference in microbiota composition may emerge from both its species and strain constituents. It is important to know the precise composition especially for the gut microbiota (GM), since it can contribute to the health assessment, personalized treatment, and disease prevention for individuals and groups (cohorts). The existing methods for species and strain composition in microbiota are not always precise and usually not so easy to use. Probiotic bacteria of the genus *Bifidobacterium* and *Lactobacillus* make an essential component of human GM. Previously we have shown that in certain *Bifidobacterium* and *Lactobacillus* species the RelBE and MazEF superfamily of toxin-antitoxin (TA) systems may be used as functional biomarkers to differentiate these groups of bacteria at the species and strain levels. We have composed a database of TA genes of these superfamily specific for all lactobacilli and bifidobacteria species with complete genome sequence and confirmed that in all *Lactobacillus* and *Bifidobacterium* species TA gene composition is species and strain specific. To analyze composition of species and strains of two bacteria genera, *Bifidobacterium* and *Lactobacillus*, in human GM we developed TAGMA (toxin antitoxin genes for metagenomes analyses) software based on polymorphism in TA genes. TAGMA was tested on gut metagenomic samples. The results of our analysis have shown that TAGMA can be used to characterize species and strains of *Lactobacillus* and *Bifidobacterium* in metagenomes.

# BACKGROUND

The human gastrointestinal tract is a habitat for a wide variety of microorganisms, mainly bacteria (*Montiel-Castro et al., 2013*; *Thakur et al., 2014*). Human gut microbiota (GM)

consists of $10^{14}$ cells of more than 1,000 species (*Browne et al., 2016*; *Rajilić-Stojanović & De Vos, 2014*). The GM has been described as a new endocrine organ that plays an important role in formation and maintenance of immunity and overall homeostasis, including formation of neuropsychological and behavioral features (*Flint et al., 2012*). An individual metagenome depends on the country, urban or rural dwelling, age group, diet preferences, and general health state or various diseases (*Lloyd-Price, Abu-Ali & Huttenhower, 2016*). The GM of healthy adults (from 2.5 to 65 years) is quite stable (*Faith et al., 2013*). Individual differences of GM and their effect on the macroorganism depend on the microbiome composition not only at the level of phyla or species but also at the strain level. Subspecies variates of bacterial strains can display a substantial variation in metabolism type or other properties (*Zhu et al., 2015*; *Greenblum, Carr & Borenstein, 2015*). Recently it has been reported that the presence of particular strains of the same species correlates with the onset and progression of human disorders including adiposity and insulin resistance (*Zhang & Zhao, 2016*). It is important to know the precise composition especially for the GM, since it can contribute to health assessment, personalized treatment, and disease prevention for individuals and groups (cohorts). However, human microbiota composition is usually analyzed at the family or genus level rather than at the species and strains level. Methods for strain characterization in microbiota are still not adequate. Most of the software tools based on the presence of marker genes like MetaPhlAn (*Segata et al., 2012*) or MG-RAST (*Keegan, Glass & Meyer, 2016*) have resolution at most at the species level. Approaches based on gene copy number variation (*Greenblum, Carr & Borenstein, 2015*) require very deep metagenome sequencing (>500×). We believe that identification of novel functional markers highly represented in most human gut microbiome samples and using them to characterize the bacteria of the GM at the strain level would make an important contribution into analysis of metagenome samples.

Toxin-antitoxin gene systems (TASs) are present in the genomes of the overwhelming majority of bacteria and archaea (*Unterholzner, Poppenberger & Rozhon, 2013*; *Klimina, Poluektova & Danilenko, 2017*). They are involved in bacterial persistence, antibiotic tolerance, stress response, apoptosis, biofilm formation (*Van Melderen, 2010*; *Hu, Benedik & Wood, 2012*). Type II TASs are most numerous and well-studied. They usually consist of two components, the toxin (T) and antitoxin (A). Usually T and A genes are located nearby and form an operon. The toxin causes death of cells or suppresses their proliferation; the antitoxin interacts with the toxin blocking its activity. Toxins target mRNAs, ribosomes, DNA-gyrases, cytoskeletal proteins and cell wall synthesis systems. The antitoxin is less stable than the toxin. Under-suppression of transcription or translation the antitoxin is digested by proteases to release free toxins leading to cell death or growth arrest (*Yamaguchi, Park & Inouye, 2011*). Previously, we showed that genes of the MazEF and RelBE superfamilies of type II TAS were present in all tested strains of several species of *Bifidobacterium* and *Lactobacillus* and demonstrated how TA genes could be used to identify species and strains in these bacteria (*Klimina et al., 2013*; *Averina et al., 2013*; *Krügel et al., 2015*).

*Bifidobacterium* and especially *Lactobacillus* make a small part of human gut microbiome. Even so, they are cultivated and well-studied bacteria that have health-promoting properties and contribute to homeostasis in the host (*Turroni, Van Sinderen & Ventura, 2011*; *Walter, 2008*). It has been reported that individual gut microbiomes include strains of *Bifidobacterium* and *Lactobacillus* specific for the individual, which agrees with the general pattern of individual strain preference observed for many bacterial taxa (*Derrien & Van Hylckama Vlieg, 2015*; *Saez-Lara et al., 2015*). Bacterial strains administered to animals (rodents) and humans substantially influenced the microbiota composition, as well as metabolomic and immunity processes, ultimately affecting disease development. The effect substantially depended on the particular bacterial strain being administered, including *Bifidobacterium* and *Lactobacillus* strains (*Derrien & Van Hylckama Vlieg, 2015*; *Saez-Lara et al., 2015*).

We suppose that TASs can provide additional functional markers for metagenomic analysis of species and strain diversity of the genera *Lactobacillus* and *Bifidobacterium* (*Klimina et al., 2018*). To this end, we created a database of MazEF and RelBE chromosomal T and A genes in all complete genomes of *Bifidobacterium* and *Lactobacillus* genus and TAGMA software which conducts species and strain identification. We tested TAGMA to identify species and strains of lactobacilli and bifidobacteria in 147 metagenome samples from the Human Microbiome Project (available in the Human Microbiome Project database (https://www.hmpdacc.org/HMASM/, subtab "stool")) as well as in five in-house samples (see Methods: In-house metagenome characterization). The results were compared with those obtained with other programs, PhymmBL and MetaPhlAn. Based on a limited number of well selected markers TAGMA displays performance at least comparable to that displayed by those programs, may be somewhat underperforming PhymmBL. TAGMA can identify species of lactobacilli and bifidobacteria displaying prediction quality comparable to that of PhymmBL and MetaPhlAn but it is based on the small number of carefully selected markers, which thus can be obtained with very deep targeted sequencing, and also it works much faster than PhymmBL or MetaPhlAn. In some cases TAGMA also can identify individual bacterial strains, an option which is not implemented in MetaPhlAn or PhymmBL.

## METHODS

### Software for metagenomic analysis

Toxin antitoxin genes for metagenomes analyses is a pipeline consisting of existing published software and in-house scripts (https://github.com/LabGenMO/TAGMA). In the first step the algorithm scans BLASTN alignments of TAS and identifies markers (substitutions and indels) that distinguished gene variants, identified by all to all BLASTN alignments. In the second step metagenomic reads are aligned with TAS genes using BowTie2 (*Langmead & Salzberg, 2012*) algorithm. In the third step the aligned reads that support particular marker variants are identified.

Since some marker position can be not covered by reads, while some other could appear due to sequencing errors, the discernibility matrix $G(g \times g)$ is constructed in the fourth stage. Here, $g$ is the set of detected genes or gene variants, $G(1, 2) = 1$ means that gene

1 cannot be distinguished from gene 2 (0 otherwise). Due to fragmental read coverage some genes that are theoretically distinguishable by the complete marker set become indistinguishable with the observed marker set. The TAGMA reports such cases and outputs the smallest possible group of genomes, that can still be distinguished with the observed set of markers (Fig. 1).

In the fifth step St($s \times s$), the strain discernibility matrix is build. Here, $s$ is the number of strains that have at least one detected genetic marker. For instance, St(3,4) = 1 means that strain number 3 is not distinguishable from strain number 4. This matrix is not symmetric. One cannot distinguish strain 1 from strain 2 if the coverage of all detected genes in strain 1 is not enough to distinguish variants of these genes from those in strain 2 (or if these genes are completely identical). But there are cases when strain 2 can be distinguished from strain 1 if strain 2 contains another set of detected genes or at least one gene variant that has a characteristic marker position. In this case one is not allowed to consider strain 1 as a false positive, because the genomes in metagenomic samples are not fully covered with reads and information can be missing due to sequencing depth deficiency.

In the sixth step, sets of indiscernible strains are derived from the strain discernibility matrix. The number of strains that cannot be distinguished from each other and from the target strain is used as the measure of performance. This measure is lower for better detected strains.

Toxin antitoxin genes for metagenomes analyses can be used for identification of *Lactobacillus* and *Bifidobacterium* species and individual strains in metagenomes.

## Collection and annotation of TAS type II genes in *Lactobacillus* and *Bifidobacterium* known genes

First, we have collected a compilation of known genes of listed TASs type II superfamilies MazEF and RelBE in *Lactobacillus rhamnosus, L. casei, L. helveticus, L. plantarum, Bifidobacterium longum* (from NCBI Gene database (http://www.ncbi.nlm.nih.gov/gene)) and used it for gene annotation. Then for extended annotation of T and A genes we used the *Lactobacillus* and *Bifidobacterium* strains with complete genome sequence. We identified homologous regions in all complete genomes of *Bifidobacterium* (53 complete genomes) and *Lactobacillus* (72 complete genomes) (data available in NCBI (http://www.ncbi.nlm.nih.gov/genome)), located open reading frames and checked for proteins corresponding to annotated genes. Homologous sequences were identified with TBLASTX aligner with *e*-value threshold $10^{-20}$, protein coding reading frames were predicted with GeneMarkS algorithm (http://exon.gatech.edu/GeneMark/) (*Besemer, Lomsadze & Borodovsky, 2001*), whereas protein sequences were assessed with InterPro (http://www.ebi.ac.uk/interpro/) (*Jones et al., 2014*). For target regions we adopted those in which homologous sequence overlapped with gene predicted by GeneMakrS for more than 80%. We considered only TASs located at chromosomes only, not at plasmids. All gene locus tags (748) are available at https://github.com/LabGenMO/TAGMA.

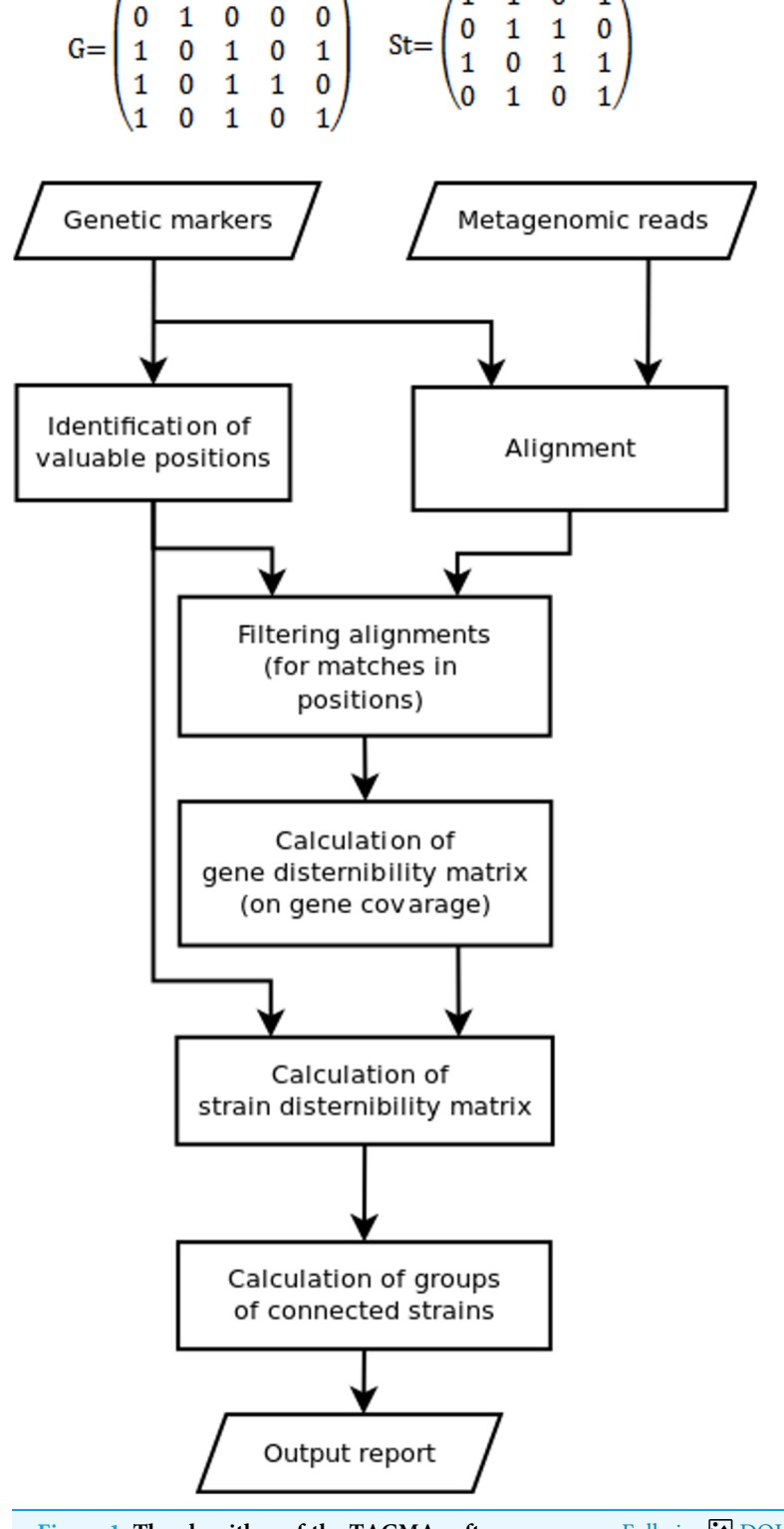

**Figure 1  The algorithm of the TAGMA software.**   

## Heatmap construction

We used BLASTN to classify all genes into groups. Genes from the same group were considered as identical if they were similar for more than 88%. Each group of genes had a set of SNPs distinguishing it from representatives of other groups. A total of 49,353 nucleotide sequences of plasmids were downloaded from NCBI by using entrez query "NUCCORE:plasmid[Title]." Nucleotide sequences of TASs genes were mapped by BLASTN program at the plasmid sequences. The hits were filtered with identity threshold values at 80%. If a plasmid hit was located for a TASs gene hit, we assumed that this TA system gene could be transferred by plasmids. A set of Python scripts tagma.py (https://github.com/LabGenMO/TAGMA) was developed to assess the presence of genes of different groups in the selected strains and species. These groups of genes were used to construct heatmaps with the help of *seaborn* python library (https://stanford.edu/~mwaskom/software/seaborn/).

## Phylogenetic analysis of metagenomic samples

To analyze diversity of species in metagenomic samples we used two software packages for taxonomical classification: MetaPhlAn (*Segata et al., 2012*) and PhymmBL (*Brady & Salzberg, 2011*). MetaPhlAn software is based on alignment of metagenomic reads with clade-specific marker genes. We used it with the default set of parameters. In contrast to MetaPhlAn PhymmBL uses BLAST (*Altschul et al., 1990*) aligner to map reads or assembled sequences on sequences of whole bacterial genomes. To prepare the data for PhymmBL, we used metaVelvet, the de novo metagenome assembler (*Namiki et al., 2012*), with the following parameters: the kmer value 55, the minimum contig length 1,000 bp. PhymmBL software was used with default parameters.

## In-house metagenome characterization

We used the gut metagenomes (feaces), isolated from people living in the Central region of Russia. All five metagenomes have been deposited in GenBank (NCBI) under the accession no: SRX1869839 (RM1, a healthy 6 year old girl); SRX1869842 (RM2, a healthy 28 year old woman), SRX1870055 (RM3, a healthy 34 year old woman), SRX1878777 (RM4, a healthy 28 year old woman), SRX1878778 (RM5, a 51 year old man with type II diabetes mellītus, the sugar level of 8.7).

## DNA extraction and quantification

DNA was extracted from the feaces (RM1-RM5) using the QIAamp DNA Stool Mini Kit (Qiagen, Hilden, Germany) according to the manufacturer protocol. The gDNA quantity was determined on the Qubit 2.0 Fluorometer (Invitrogen, Carlsbad, CA, USA) per manufacturer instructions.

## Fragmentation of DNA

Each of the gDNA samples were fragmented by nebulization using compressed nitrogen gas, nebulizers, nebulization buffer (Illumina, San Diego, CA, USA) and glycerol (Sigma, St. Louis, MO, USA). Fragmented DNA samples were purified using the
MinElute PCR Purification Kit (Qiagen, Hilden, Germany) according to the manufacturer protocol. Nebulization was performed for the duration of 1 min with nitrogen (pressure 2.1 bar) with a total input of one mg gDNA for each sample. The final size of fragmented gDNA samples was determined on Agilent 2100 Bioanalyzer (Agilent, Santa Clara, CA, USA) per manufacturer guide and was approximately of 300–500 bp.

## Illumina library preparation and sequencing

Fragmented and cleaned DNA samples were prepared using TruSeq DNA LT Sample Prep Kit (Illumina, San Diego, CA, USA) following the TruSeq DNA Sample Preparation v2 Guide starting with the "Perform End Repair" step. In brief, fragmented DNA samples were end repaired, 3′ ends were adenylated and TruSeq adapters were ligated to the each gDNA sample. The in-Line Control DNA was added to each enzymatic reaction. Libraries were then size-selected using a 2% low range ultra-agarose gel with 1X TAE buffer run at 120 V for 120 min. The 500–600 bp fragments (of which ~120 bp were the ligated adapters) were cut out with a sterile scalpel blade for each individual sample. Each gDNA sample was purified using the MinElute Gel Extraction Kit (Qiagen, Hilden, Germany) following manufacturer instructions. Then after gel extraction, the libraries were subjected to amplification by means of PCR process according to the manufacturer protocol (10 cycles). The libraries were validated by visualization on the Agilent 2100 Bioanalyzer (Agilent, Santa Clara, CA, USA), quantified using qPCR. The libraries were then sequenced as 2 × 250 bp paired-end runs on the Illumina MiSeq Systems (Illumina, San Diego, CA, USA) according to the manufacturer instruction. Raw sequencing reads were obtained using Illumina analysis software (MiSeq Reporter; Illumina, San Diego, CA, USA). Trimming v0.3 program (*Bolger, Lohse & Usadel, 2014*) was used for trimming, FastQC v.0.10.1 (*Ramirez-Gonzalez et al., 2013*) program was used for quality control. Chimeric reads where filtered with Uchime (*Edgar et al., 2011*) algorithm (which is a part of Userach v7.0 program). Classification of bacteria was made with RDP MultiClassifier v1.1 program (*Wang et al., 2007*).

# RESULTS

## Distribution of TASs genes in *Bifidobacterium* and *Lactobacillus* sp

Previously, we showed that different *L. rhamnosus* and *L.fermentum* strains have different sets of TASs from RelBE and MazEF superfamilies. The same was shown for genus *Bifidobacterium*. Primary investigation showed that genes of these families are found in all studied *Lactobacillus* and *Bifidobacterium* species (*Klimina et al., 2013*; *Averina et al., 2013*; *Poluektova et al., 2017*). Now we investigated how the TASs of RelBE and MazEF superfamilies are distributed in other species of *Lactobacillus* and *Bifidobacterium* stored in the GenBank. For this purpose, genes of MazEF and RelBE superfamilies were annotated (see Methods) and a database of T and A genes has been constructed. The same TA genes may be present on the chromosome as well as, rather rare, on a plasmid (*Remisetti & Santhos, 2016*). Our objective was to create a database containing only the chromosomal TA genes. For this purpose, we identified plasmid regions similar to TA genes located in genomes. From all TA genes 14 were found to have plasmid

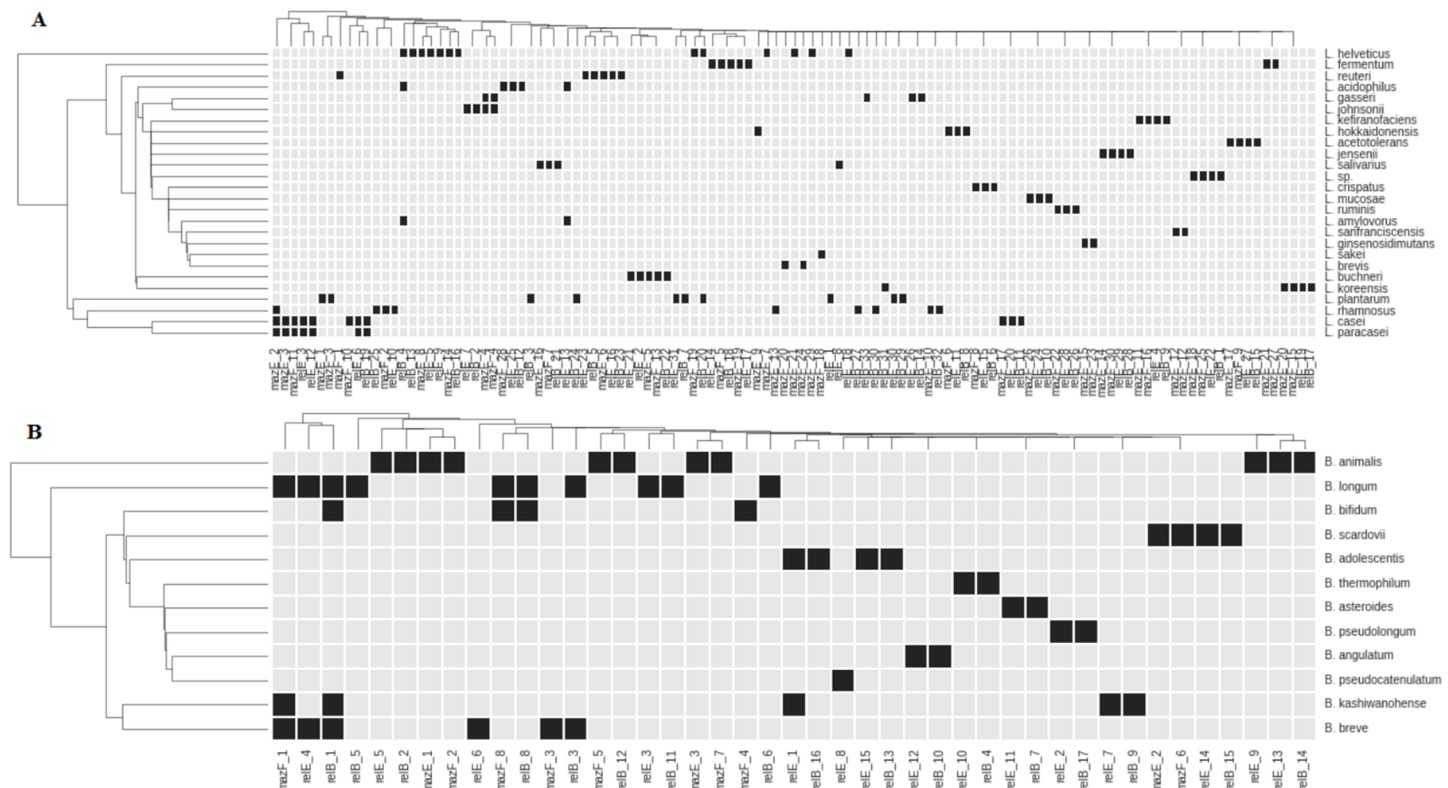

**Figure 2 Representation of TASs type II genes of superfamilies RelBE and MazEF in the species of *Lactobacillus* (A) and *Bifidobacterium* (B).** Black boxes show the presence of a gene. The name of each group of TA genes consists of the name of a gene and the number of a group.

homologs and have been removed from our database. Figure 2 displays the dot matrix showing the presence or absence of the TA genes in the species of *Lactobacillus* and *Bifidobacterium*.

All studied species, except *L. paracasei*, have species-specific genes. These genes in *L. paracasei* are the same as in *L. casei*. These species are very close and difficult to differentiate from each other (*Wuyts et al., 2017*). Distribution of almost any gene reveals some degree of species specificity (Fig. 2). In some cases, closely related species of *Lactobacillus* (*L. johnsonii*—*L. gasseri*; *L. helveticus*—*L. acidophilus*—*L. amylovorus*) share common genes (Fig. 2A). Several common genes have also been found in some bifidobacteria species: *B. longum, B. bifidum, B. breve, B. adolescentis* and *B. kashiwanohense* (Fig. 2B). *B. longum* and *B. breve* belong to the "*B. longum* group," *B. adolescentis* and *B. kashiwanohense* belong to the "*B. adolescentis* group" and *B. bifidum* belong to the "*B. bifidum* group." These species groups are always closely located in the phylogenetic tree and are more remote from other groups of bifidobacteria, indicating their common separation from other groups, and their subsequent divergence. Nevertheless, each species has its own specific set of TA genes.

We performed similar analysis for strains. Distribution of T and A genes is markedly strain specific (see Fig. S1). Strains belonging to the same species of bacteria have similar but not identical sets of T and A genes. Consequently, these genes can be used to

identify species and strains of *Lactobacillus* and *Bifidobacterium*. This property can be used to characterize the presence of specific strains of *Lactobacillus* and *Bifidobacterium* species in human microbiota.

## The software

The general idea of our approach is the same as in other techniques of metagenomic profiling (16sRNA, MetaPhlAn, PhymmBL). Each strain is associated with the specific set of genetic markers. We use two types of markers: (i) TA genes themselves (each strain has different set of TA genes, so particular TA genes found in the sample help to identify the strain); (ii) differences in the sequences of particular TA genes (each strain has pattern of point variants and indels in its TA genes).

One of the main difficulties of metagenomic analysis is the fragmented structure of genomic data. Only parts of some genes are present in the sample reads; reads for other genes can be totally missing. These difficulties may be counteracted by deeper sequencing of a small set of selected markers. An example of such marker is ribosomal RNA genes, which is efficient for species identification, although it is not always possible to distinguish between similar species but is less efficient for identification of strains. Thus, a broader set of markers is necessary for strain identification, which can include makers either present or missing in different strains, which thus can be identified by marker combinations. Additionally, the target sequences can be variable enough, to use specific single nucleotide for strain characterization. Both properties are valid for genes belonging TAS families: these systems may be present of absent in different strain genomes and their genes are not very conservative.

Our approach to metagenomic analysis consists in two stages. In the first stage all known genomes of the studied subgroup are compared to evaluate possible variable TAS-related genetic markers, and assess occurrence of different variants in the strains. At this stage all sequences are aligned against each other (all-over-all blast) and the differences between genome sequences are identified. Genetic markers thus become grouped into homogeneous groups and intragene variants (point substitutions and indels) that distinguish genomic sequences in each group. The metagenome is analyzed in the second stage. All reads are mapped against genetic markers with Bowtie2 program. Algorithm tracks mismatches that occur while alignment metagenomic reads and genetic markers, if mismatch occurs in the position that distinguishes two homogeneous genetic markers, such marker is discarded. To allow for the fragmented structure of metagenome coverage with the reads we considered possible marker dropouts due to low read coverage. The entire pipeline is called TAGMA and is available at (https://github.com/LabGenMO/TAGMA).

## Metagenomic analysis

To test our software, we studied representation of *Lactobacillus* and *Bifidobacterium* species in metagenomes. First, we analyzed five metagenomes, RM1-RM5 studied in our Lab and obtained from people of different age and state of health (see Methods). The faecal samples has been collected in the identical conditions, DNA was extracted

**Table 1 The presence of genera *Bifidobacterium* and *Lactobacillus* in the metagenomes determined by MetaPhLAn2 and PhymmBL programs.**

|  | RM1 | | RM2 | | RM3 | | RM4 | | RM5 | |
|---|---|---|---|---|---|---|---|---|---|---|
|  | Ph | Mt | Ph | Mt | Ph | Mt | Ph | Mt | Ph | Mt |
| *Lactobacillus sp* | 0.1 | – | 1.32 | – | 0.17 | – | 0.92 | – | 2.33 | 2.66 |
| *Bifidobacterium sp* | 1.61 | 5.17 | 2.79 | 12.77 | 8.31 | 23.63 | 1.77 | 10.47 | 2.43 | 5.74 |

**Note:**

Ph, PhymmBL; Mt, MetaPhLAn2.

and libraries have been prepared. Both MetaPhlAn2 and PhymmBL indicated that all samples contained bacteria of *Lactobacillus* and *Bifidobacterium* genera (Table 1). To validate that *Lactobacillus* and *Bifidobacterium* species were present in the microbiomes we plated feces suspension on nutrient media (K. Klimina and E. Poluektova, 2018, unpublished data). *Lactobacilli* was found in RM1 in much lesser quantities than in other metagenomes, which is the known feature of children microflora (Botina et al., 2010).

We applied TAGMA for analysis of five in-house human gut metagenomes up to the level of species; the results were compared with those obtained using MetaPhLAn2 and PhymmBL (Table 2). PhymmBL uses BLAST to align sequences of complete metagenomes against genomes of prokaryotes. MetaPhLAn2 performs read clustering before doing BLAST searches, thus increasing time efficiency. The analysis showed that those species of *Bifidobacterium,* which were detected with both PhymmBL and MetaPhLAn2, were discovered by TAGMA. *Lactobacillus* genus was present in very low quantities (Table 1) and it was difficult to detect the corresponding species. Those species of *Bifidobacterium* and *Lactobacillus* which were not detected by TAGMA were either species that were present in metagenome in small amounts (less than 0.4%) or that had no markers with sufficient read coverage.

Table 2 illustrates that the TAGMA gives more information than MetaPhLAn2 but less than PhymmBL. But the advantage of TAGMA is that it can analyze metagenome up to strain level (Table S1 and S2) if the metagenome contains a specific set of TA genes or at least to the level of a group of strains if these strains contain identical T and A genes (see Table S1 and S2).

Then we tested TAGMA on the open access data. For the analysis, we selected 147 samples of the intestinal metagenomes of healthy human subjects, available in the Human Microbiome Project database (https://www.hmpdacc.org/HMASM/, subtab "stool"). These samples were isolated from the feces of healthy men and women aged 18–40 years old living in the United States. Previously, Kovtun et al. analyzed these 147 samples. In their analysis, they searched the metagenomes for bacterial enzymes involved in the synthesis of neuroactive compounds and determined the taxonomic composition using both Kraken and MetaPhLAn2 programs. Their analysis confirmed the presence of lactobacilli and bifidobacteria in the metagenomes (at the species level) (Kovtun et al., 2018). Using TAGMA, we identified *Lactobacillus* and *Bifidobacterium* in each metagenome down to the strain level (Table S3). Table S3 shows the strains (or groups of strains) of *Lactobacillus* and *Bifidobacterium* that satisfied the following conditions: the

Table 2 The presence *Bifidobacterium sp.* and *Lactobacillus sp.* in five in-house metagenomes detected by three different programs.

| Species | PhymmBL | MetaPhLAn2 | TAGMA |
|---|---|---|---|
| **RM1** | | | |
| *B. adolescentis* | + | + | + |
| *B. bifidum* | + | + | + |
| *B. longum* | + | + | + |
| *B. breve* | − | − | + |
| *L. rhamnosus* | − | − | + |
| **RM2** | | | |
| *B. adolescentis* | + | + | + |
| *B. angulatum* | − | + | + |
| *B. bifidum* | + | + | + |
| *B. longum* | + | + | + |
| *B. breve* | + | − | + |
| *B. dentium* | + | − | − |
| *L. delbrueckii* | + | − | − |
| *L. fermentum* | + | − | − |
| *L. salivarius* | + | − | − |
| **RM3** | | | |
| *B. adolescentis* | + | + | + |
| *B. bifidum* | + | − | − |
| *B. breve* | + | − | + |
| *B. catenulatum* | − | + | − |
| *B. dentium* | + | − | − |
| *B. longum* | + | + | + |
| *B. pseudocatenulatum* | − | + | + |
| *B. kashiwanohense* | − | − | + |
| **RM4** | | | |
| *B. adolescentis* | + | + | + |
| *B. bifidum* | + | + | + |
| *B. longum* | + | + | + |
| *L. fermentum* | + | − | − |
| *L. rhamnosus* | − | − | + |
| **RM5** | | | |
| *B. adolescentis* | + | + | + |
| *B. animalis* | + | − | − |
| *B. bifidum* | + | − | − |
| *L. ruminis* | + | + | + |
| *L. fermentum* | + | − | |

TA genus coverage is more than 60% and the number of markers detected in a strain more than two (or 50%). The composition and abundance of *Lactobacillus* and *Bifidobacterium* strains were significantly different between all the metagenomes.

## DISCUSSION

Two main approaches are currently used to characterize taxonomic diversity in metagenome samples: the whole metagenome shotgun analysis and analysis of PCR amplicons from the ribosomal 16S RNA gene (*Jovel et al., 2016*). Lactobacilli and bifidobacteria constitute only a small part of human GM (0.5%, and 10%) (*Zhernakova et al., 2016*), therefore the existing methods not always can adequately assess the presence of these genera (especially lactobacilli). Identification of species and strains in microbiota is even more difficult. Yet the results of metagenome analysis at this level make a useful characteristic of individual microbiota.

In this report, we show that genes of type II TASs can be used as functional markers for computer assisted species and strains characterization of lactobacilli and bifidobacteria in human GM. The database of T and A genes for these two genera of bacteria has been created and it has been shown that distribution of TAS is species- and strain specific.

The program TAGMA based on the TASs of *Lactobacillus* and *Bifidobacterium* has been developed and tested on five metagenomic samples. It turned out that TAGMA could effectively identify *Lactobacillus* and *Bifidobacterium* species and sometimes their individual strains in the metagenome; in some cases, TAGMA identified specific groups of strains. TAGMA can be used for characterization of individual metagenomes or groups of metagenomes (e.g., people from different habitats). TAGMA was compared with existing methods of metagenomic analysis (MetaPhLAn2, PhymmBL), and at the species level outperformed MetaPhLAn2 but proved to be less sensitive than PhymmBL. This difference can be explained by the fact that each program uses different markers to identify the genus and species of bacteria. The most comprehensive information of species composition in metagenomes is revealed using several programs.

## CONCLUSION

In contrast to other programs, based on a large set of genomic markers (up to one million; https://bitbucket.org/biobakery/metaphlan2), TAGMA employs only a small set of genes (from 2 to 10) to determine species in metagenomes. Therefore, TAGMA is much more time efficient as it only takes several hours to analyze one metagenome, whereas MetaPhLAn2 and PhymmBL analysis takes several days on a desktop computer with only one computational core. The developed software can also be applied for metagenomic analysis of the oral and vaginal cavity, where lactobacilli are dominating. We plan to expand the TASs database not only for analysis of the strain diversity of the *Lactobacillus* and *Bifidobacterium* species but also for other bacteria found in the human gastrointestinal tract.

## ACKNOWLEDGEMENTS

This work was performed using the sequencing equipment of EIMB RAS "Genome" center (http://www.eimb.ru/rus/ckp/ccu_genome_c.php) and using the computational resources of VIGG RAS «Genetic Polymorphisms» Center (http://vigg.ru/institute/ckp/ckp-obn-ran-geneticheskii-polimorfizm/).

### Funding

Work in the reported study such as creating database TASs, analysis and data processing and developing the program was funded by RFBR according to the research project № 18-34-00011. Sequencing and following bioinformatics analysis of bacterial communities in human gut were funded by the Russian Science Foundation grant N 14–15-01083. Vsevolod Makeev received support from the Presidium of Russian Academy of Sciences Program for Basic Research in Molecular and Cell Biology and Post Genome Technologies #18. The funders had no role in study design, data collection and analysis, decision to publish, or preparation of the manuscript.

### Grant Disclosures

The following grant information was disclosed by the authors:
RFBR according to the research project: 18-34-00011.
Russian Science Foundation: 14–15-01083.
Presidium of Russian Academy of Sciences Program for Basic Research in Molecular and Cell Biology and Post Genome Technologies #18.

### Competing Interests

Vsevolod J. Makeev is an Academic Editor for PeerJ.

### Author Contributions

- Ksenia M. Klimina conceived and designed the experiments, performed the experiments, analyzed the data, contributed reagents/materials/analysis tools, prepared figures and/or tables, authored or reviewed drafts of the paper, approved the final draft, provided access to the VIGG RAS «Genetic Polymorphisms» Center.
- Artem S. Kasianov conceived and designed the experiments, performed the experiments, analyzed the data, prepared figures and/or tables, authored or reviewed drafts of the paper, approved the final draft.
- Elena U. Poluektova analyzed the data, authored or reviewed drafts of the paper, approved the final draft.
- Kirill V. Emelyanov performed the experiments, analyzed the data, prepared figures and/or tables.
- Viktoriya N. Voroshilova performed the experiments, analyzed the data, prepared figures and/or tables.
- Natalia V. Zakharevich analyzed the data.
- Anna V. Kudryavtseva performed the experiments, analyzed the data, contributed reagents/materials/analysis tools, provided access to the EIMB RAS "Genome" Center.
- Vsevolod J. Makeev conceived and designed the experiments, performed the experiments, contributed reagents/materials/analysis tools, authored or reviewed drafts of the paper, approved the final draft.

- Valery N. Danilenko conceived and designed the experiments, contributed reagents/ materials/analysis tools, authored or reviewed drafts of the paper, approved the final draft.

## Data Availability

Sequence data (FASTQ files) (metagenomic sequencing data of the five Russian samples) were deposited in the NCBI's SRA (Bioproject: PRJNA324672).

We released the source code of the program in GitHub: https://github.com/LabGenMO/TAGMA.

## Supplemental Information

Supplemental information for this article can be found online at http://dx.doi.org/10.7717/peerj.6554#supplemental-information.

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
