# Peer review of "Employing toxin-antitoxin genome markers for identification of Bifidobacterium and Lactobacillus strains in human metagenomes"

_PeerJ, doi:10.7717/peerj.6554_

## Round 0.1 · original submission · Major Revisions

Please consider critical remarks on the manuscript. We had two reviews and can't wait another one. The paper needs revision, I believe not really so major. If you need more time for revision, please take it.

·

Basic reporting

Klimina et al report use of toxin-antitoxin system (TAS) genes in metagenomes as a tool for strain identification in Lactobacillus and Bifidobacterium genera. For this purpose authors created a software program which was compared with existing programs. Using these tools authors report prevalence of different Lactobacillus and Bifidobacterium strains in database available metagenomes from different geographic locations including Russia, USA, Denmark and Japan. They also used this approach to analyze metagenomes from 5 fecal human samples. The results validate the new program and TAS approach as a useful tool for identifying Lactobacillus and Bifidobacterium strains.

Experimental design

no comments

Validity of the findings

no comments

Additional comments

Specific comments
1. The rational of analyzing metagenomes from fecal samples in 5 individuals is not clear. In fact, for the purpose of this study, analysis of database available metagenome data is sufficient. However, whether the authors wanted to validate their in silico analysis with in vitro identification of Lactobacillus and Bifidobacterium strains having the fecal samples would be an advantage. Please clarify the rational of using fecal samples.
2. It is not clear what were the criteria for selection of 4-5 metagenome data from 4 different geographic locations? Were these sex- and age matched? I think that analyzing only 5 individuals are not sufficiently representative to draw the conclusion about the prevalence of different strains of Lactobacillus and Bifidobacterium in different countries. Please state the limitations.
3. Please explain if the TAS method can be used for quantification of Lactobacillus and Bifidobacterium strains in metagenomes and whether the results are depended on the bacterial growth cycle i.e. can be influenced by lack of standardization in fecal sampling procedure.
4. Authors stated that ethical approval and consent to participate were not applicable to this study. What about use of fecal samples from 5 individuals?
5. The paper should be better structured to emphasize the objectives and to clearly present the data.
6. English language should be improved, including rephrasing of several sentences. Ex, line 299-300 sentence can be replaced simply by "country-specific".

·

Basic reporting

1. The software tool (TAGMA) was written to allow the users to perform analysis of metagenomic samples with strain resolution. This application might be of interest for a quite broad audience, however a number of issues listed below make the tool difficult and even dangerous to use in its current form:
- Code safety: the current implementation lacks error and exception handling (e.g try-except statements). Unit test should be also included (ideally for every one of the 21 functions implemented), to ensure that the software deals properly with edge cases, non-standard inputs, etc.
- Test data: Please provide a simple example of test data set. Please also provide a test data set when non TA-based markers could be used.
- Dependencies: the TAGMA software tool relies on a number of external dependencies (BLAST, Bowtie. etc), however the repository lacks information on how those dependencies should be installed and configured; how the process should be changed to accommodate Windows and Mac users. Please provide these additional instructions.
- Resources: Please list computational resources required for the tool (RAM, number of CPU). Could it be adopted for use on HPCs? What are the limitations? Which steps of the pipeline can benefit from additional computational resources available? Please also include time estimates for a typical dataset to be analysed.
- Documentation: In the documentation the main program is referred as (‘toximet.py’). However, the repository contains just one python file with a different name (‘bt2_snp_11.py’).
- Documentation: Generally, the documentation is very confusing and should be improved considerably. Currently the GitHub repository contains a README file with some background information on how to use the tool, but poor formatting makes it a struggle to read through: This issue could be solved using appropriate markdown syntax and GitHub Pages.

2. The English language of the paper should be improved to ensure that an international audience can clearly understand the text. The text might benefit from a proof-reading by a native speaker. Some examples, where the language can be changed include Lines 39-41; 59-60; 79-82; 221-222; 225-227; 229-230; 236-237; 242; 250; 267-268; 276-278; 293; 315; 317, etc.

3. The authors mention that they have created a database of toxin and antitoxin genes for Lactobacillus and Bifidobacterium genera (Lines 31-32, 83, 117, etc), however this database has not been made available for the potential readers. It is not completely clear, are the authors referring to a blast database, a list of gene locus tags or is it more than that? In the latter case, the database might be of interest for a broad audience, so some sort of an interface should be provided.

Experimental design

1. Most importantly, the focus of the manuscript seems to be vague. Currently the manuscript balances between a description of a bioinformatics software tool to perform strain profiling in metagenomic data and a study of Bifidobacteria and Lactobacillus TA II systems occurring in a few human metagenomic data sets. If the software is made the primary focus, the quality of the code and documentation should be improved dramatically, to ensure that the tool receives enough trust to be used by a broader audience. If a study of TA II systems is made a priority, than considerably more data sets, including a wider selection of publicly available human gut metagenomes should be analysed.
2. Provided analysis in the paper considers TAS II systems in two bacterial genera: Bifidobacterium and Lactobacillus. Hence, the title of the manuscript is misleading, in its current form it implies much broader use of the software. If a broader use is indeed plausible, the authors should provide additional evidence that the TAGMA software can be used for other genera of bacteria and archaea. Otherwise, this claim should be reduced and the title of the manuscript - revised.

Validity of the findings

1. The conclusions the authors are making when comparing TAGMA performance to the existing software tools are not quantitative enough. Some examples include:
Lines 32-34 (‘It was tested on metagenomic samples and shown that it could be used for identification of Lactobacillus and Bifidobacterium species and sometimes for identification of individual strains in the metagenome.)
Lines 324 - 326: (‘TAGMA was compared with existing methods of metagenomic analysis (MetaPhlan, PhymmBL), and at the species level outperformed MetaPhlan2 but proved to be somewhat less sensitive than PhymmBL’)
I think ‘sometimes’ and ‘somewhat’ are not quantitative enough here. Please clearly state limitations of the software along with its sensitivity and specificity.

2. Please also see the comment #1 from the basic reporting section on the code and documentation quality.

Additional comments

In addition, please see more specific comments below:

Abstract:
Background: Please clearly state that the manuscript addresses human/animal type gut microbiomes. Bifidobacteria and Lactobacillus are not the common component of termite gut microbiomes for instance (Lines 16-18).
Results: I think the ‘results’ part of the abstract lacks sufficient details. It would be great to include types of metagenomic samples, how many of those? Are they publicly available or sequences specifically for this study?
Results: Please add a brief description of what the mentioned programmes are used for (PhymmBL and MetaPhlAn) and their main implementation principles.
Results: Please add a brief description of the principle of marker selection by TAGMA in a nutshell. Currently, it is not clear at all.
Conclusions: Please provide the rationale to focus of RelBE and MazEF superfamilies.
Conclusions: Lines 32-34 (‘It was tested on metagenomic samples and shown that it could be used for identification of Lactobacillus and Bifidobacterium species and sometimes for identification of individual strains in the metagenome.) I think ‘sometimes’ is not quantitative enough here. Please clearly state limitations of the software along with its sensitivity and specificity.

Background:
Lines 47-48: ‘According to their phylum composition gut metagenomes can be classified into three enterotypes [6]. ’ This statement has been challenged recently (Knights et al 2014; Gorvitovskaia et al 2016 ), and the concept of ‘enterotype’ is still not consolidated so this might be worth mentioning.
Lines 51-52: ‘Human microbiota composition has been analyzed mostly at the taxa level or in a lesser degree at the species level’. Species is also a taxon, I think the phrasing is confusing here and should be improved.
Lines 63-65: Is there any evidence that strain variation of Bifidobacterium and Lactobacillus has any impact on human gut microbiome and human health in general?
Lines 82-83: please include the name of the software tool.
Lines 84-85: please include the number of metagenomic data sets analysed.
It would be great to add additional 1-2 sentences to summarise benefits of the TAGMA software tool compared to the existing solutions.

Implementation
Please add a general sentence/paragraph to summarise what the TAGMA software is doing.
Please also list the dependencies and reference installation instructions.
Lines 89-91: not immediately clear how markers are identified and what is a definition of a marker in this specific context. Please rephrase.
Since this section is used to describe the software, there must be a link to the source code.


Methods:
Lines 139-140: please make the referred Python scripts available for the readers.
Lines 187-188: please provide a reference for the Trimming v.0.3 software tool.

Results:
Line 214: please rephrase, ‘slightly less pronounced’ sound very vague.
Lines 229-234: Please clearly state how the TAGMA approach is different from the other mentioned ones.
Lines 240-241: Please include an estimate of gene conservation in the TAS genes. What exactly does it mean (‘not very conservative’);
Lines 252-253: Please include the coverage threshold used to filter out the markers.
Line 264: Please add a reference supporting this statement.

Figures and tables:
Figure 1: Please add the name of the software in the figure legend title.
Figure 1: Please explain what do you mean by ‘valuable positions’.
Figure 3: Please replace ‘Japanese’ with ‘Japan’ in the figure legend.
Table 1: It is not clear what the values in the table mean. Please include additional explanation in the table title or header.

Some additional comments:

Please convert supplementary tables in a csv format to make them easier for readers to reuse and interact with.
GM is used as an abbreviation for gut microbiota too often. Please use ‘gut microbiota’ to improve readability of the text.

---

## Round 0.2 · Major Revisions

Though we have only one re-review, the manuscript still needs revision. Second reviewer had only minor remarks. I believe current reviewing remarks have rather technical character, you'd fix it soon. Encourage you re-submit revision earlier then standard 55 days. Today is 1 January. Good start for science. Happy New Year!

·

Basic reporting

- Please proofread the manuscript and the documentation, English language could still be improved, for some examples please see general comments.

- Lines 206-207: ‘Primary investigation showed that genes of these families are found in all studied Lactobacillus and Bifidobacterium species.’ Please provide evidence or reference for this statement.

- Lines 140-141:
Please add metadata for the files containing toxin and antitoxin genes to the github README. Metadata should contain brief description on how those were generated. Also, in this case, since these are just several small fasta files, the term ‘database’ might be misleading, I would recommend to change the term to ‘list of genes’, ‘collection of genes’ or something similar.

- Lines 150-151: It is not clear to which exactly python scripts authors are referring here, is it the main ‘toximet.py’ script or something else?

Experimental design

-

Validity of the findings

-

Additional comments

1). Abstract:
Line 16: ‘Recent research has indicated that each individual may have a unique genotype.’
The statement is too broad, reflects common knowledge which is not so recent. This sentence is also disjoint from the following one. I would recommend starting directly with microbiomes.
Line 21: please spell out ‘TA’, this is the first mention.
Lines 21-23: ‘The objective of this work was to create a software based on polymorphism in TA genes to analyze representation of species and strains of Bifidobacterium and Lactobacillus in human gut microbiota.’
I think emphasis might be wrong here, the main objective should still be solving a defined biological problem, writing a software is just a solution. Please clearly explain the problem and state the rationale of writing the software (are there similar solutions, why they are not suited for certain applications, etc).
Lines 25-26: ‘We created database of toxins and antitoxins genes of these superfamily specific for lactobacilli and bifidobacteria’
There might be a word missing in this sentence.
Lines 29-30: ‘Based on a limited number of well selected markers TAGMA displays good time performance as compared with results of popular programs PhymmBL and MetaPhlAn.’
Please explain what is ‘well-selected marker’ and add quantitative estimation of ‘good time performance as compared with results of popular programs PhymmBL and MetaPhlAn.’

2) Background:
Line 35: there is a space missing between ‘then’ and ‘1000’ . Also, please replace ‘then’ with ‘than’.
Lines 40-41: please revise grammar.
Line 50: please add references for MetaPhlAn or MG-RAST software.
Line 51: ‘very deep metagenome sequencing’. How deep?
Line 60: Either replace ‘species specific’ with ‘species-specific’ or drop ‘species’.
Line 64: reference is missing.
Lines 82-84: please see comment #3 to the abstract.
Line 89: please provide more background information on ‘in-house’ samples.
Line 90: please briefly describe the results of the mentioned comparison.

3) Methods:
Line 93: please drop ‘basically’;
Line 94: please replace ‘inhouse’ with ‘in-house’;
Line 98: please replace ‘genes of TAS’ with ‘TAS genes’.
Line 137: please clarify what ‘translates’ mean in this context.
Lines 150-151: It is not clear to which exactly python scripts authors are referring here, is it the main ‘toximet.py’ script or something else?
Line 163: please check the spelling.

4) Results:
Lines 219-220: Please revise grammar.
Lines 222-223: Please use italics for species names.
Line 244: Please explain/ rephrase ‘good strain marker set’;
Line 249: Please revise grammar.
Line 258: Please revise grammar.
Lies 260-261: ‘The entire pipeline is called TAGMA (Toxin Antitoxin Genes for Metagenomes Analyses) and is available at [https://github.com/LabGenMO/TAGMA].’
Please rename the main python script (‘toximet.py’ to ‘tagma.py’) for consistency or explain the reasons to keep a separate name.
Line 273: Please check the spelling.
Line 279: ‘Lactobacillus genus was represented in a smaller amount’ Please, rephrase.

5) Discussion:
Line 309: please check grammar, typos and misspelled words;
Lines 310-312: please explained what ‘prolonged maintaining of strains’ means in this context.
Lines 328: Please rephrase ‘pretty large set’

---

## Round 0.3 · accepted · Accept

After two review rounds the manuscript is much improved. There are some minor typos, as indicated by reviewer 1 (see below). I believe another review round is not necessary. Just fix it, please, while in production.

# ·

Basic reporting

see general comments

Experimental design

see general comments

Validity of the findings

see general comments

Additional comments

The manuscript has been substantially revised and improved, I have few minor comments.
1. Abstract, 1st sentence is too general, start from “Each individual has a unique microbiota composition emerging from both…
2. l. 20, microbiota is not always…
3. l. 31, strains
4. l. 23, 25, 27, toxin-antitoxin system is abbreviated here as TA while in the following manuscript as TAS, please be consistent.
5. l73-75, delete “individual” at 1st place, what is “individual strain preference” is not clear preference for what?
6. l. 78 what disease?
7. l. 83, spell out TAGMA at 1st mentioning
8. l. 89, well-selected
9. l. 274, ‘non-shown data’

·

Basic reporting

I have no further comments.

Experimental design

I have no further comments.

Validity of the findings

I have no further comments.